# Image Hashing via Cross-View Code Alignment in the Age of Foundation Models

## Abstract

Efficient large-scale retrieval requires representations that are both compact and discriminative. Foundation models provide powerful visual and multimodal embeddings, but nearest neighbor search in these high-dimensional spaces is computationally expensive. Hashing offers an efficient alternative by enabling fast Hamming distance search with binary codes, yet existing approaches often rely on complex pipelines, multi-term objectives, designs specialized for a single learning paradigm, and long training times. We introduce **CroVCA** (**Cro**ss-**V**iew **C**ode **A**lignment), a simple and unified principle for learning binary codes that remain consistent across semantically aligned views. A single binary cross-entropy loss enforces alignment, while coding-rate maximization serves as an anti-collapse regularizer to promote balanced and diverse codes. To implement this, we design *HashCoder*, a lightweight MLP hashing network with a final batch normalization layer to enforce balanced codes. HashCoder can be used as a probing head on frozen embeddings or to adapt encoders efficiently via LoRA fine-tuning. Across benchmarks, CroVCA achieves state-of-the-art results in just 5 training epochs. At 16 bits, it particularly well—for instance, unsupervised hashing on COCO completes in under 2 minutes and supervised hashing on ImageNet100 in about 3 minutes—on a single GPU. These results highlight CroVCA's efficiency, adaptability, and broad applicability.

## 1 Introduction

Foundation models have reshaped representation learning across vision, language, and multimodal domains (Awais et al., 2025; Radford et al., 2021; Oquab et al., 2023; Siméoni et al., 2025). Their embeddings capture rich semantic structure and enable applications such as image retrieval, text-to-image search, and recommendation systems. Yet, these embeddings are high-dimensional, making storage and nearest-neighbor search computationally expensive. This motivates the need for compact representations that preserve semantics while enabling fast large-scale retrieval.

Hashing addresses this challenge by mapping embeddings into binary codes, allowing efficient Hamming distance search with reduced memory and computation (Luo et al., 2023). However, learning high-quality hash codes for foundation models remains difficult. Existing methods often rely on multi-stage pipelines or distillation from pretrained embeddings (Cao et al., 2017), and use multi-term objectives to approximate binarization (Li et al., 2024), enforce alignment (Jang et al., 2022), or decorrelate bits (Ma et al., 2024). These designs complicate optimization, slow convergence, and are typically specialized to a single paradigm, such as unsupervised or supervised hashing (Jang & Cho, 2021; Luo et al., 2023).

This raises a central question:

*Can a single, simple framework unify unsupervised and supervised hashing by efficiently leveraging foundation model embeddings?*

We answer this by introducing **CroVCA** (**Cro**ss-**V**iew **C**ode **A**lignment), a simple principle for learning binary codes that remain consistent across semantically aligned views. Depending on the setting, aligned views may come from data augmentations (unsupervised) or class-consistent samples (supervised). Alignment is enforced via a binary cross-entropy loss, while coding-rate maximization (Wu et al., 2025; Li et al., 2022; Tong et al., 2023) prevents collapse and promotes balanced

utilization of the Hamming space. This formulation unifies unsupervised and supervised hashing under a single objective, and can naturally extend to cross-modal scenarios, which we leave for future work.

To realize this framework, we design *HashCoder*, a lightweight MLP hashing network with a final BatchNorm layer to balance bits. HashCoder can be used as a probing head on frozen embeddings or to efficiently adapt encoders through LoRA fine-tuning (Hu et al., 2022), supporting both dataset-specific adaptation and transfer from large pretraining datasets (e.g., ImageNet-1k (Deng et al., 2009)). We provide two variants: a compact MLP for small datasets (e.g., Flickr25K, COCO, ImageNet100) and a larger one for large-scale datasets (e.g., ImageNet-1k).

Our contributions are as follows:

- We propose **CroVCA** (**Cro**ss-**V**iew **C**ode **A**lignment), a simple principle that unifies unsupervised and supervised hashing under one objective.

- We introduce **HashCoder**, a lightweight MLP hashing network with BatchNorm for balanced bit usage, available in small and large variants.

- We demonstrate **efficient adaptation** of foundation models via probing and LoRA fine-tuning, enabling dataset-specific and transfer learning scenarios.

- We achieve **state-of-the-art retrieval performance** with minimal cost—for example, unsupervised hashing on COCO in under 2 minutes and supervised hashing on ImageNet100 in about 3 minutes, trained for only 5 epochs on a single GPU.

## 2 RELATED WORKS

**Foundation models.** Large-scale pretrained encoders provide versatile visual and multimodal embeddings that serve as backbones for several tasks Awais et al. (2025). In vision, DINOv3 Siméoni et al. (2025) achieves strong performance on downstream tasks including classification, retrieval, segmentation, and depth estimation. In multimodal settings, text-image models Radford et al. (2021); Fang et al. (2023); Li et al. (2023) enable zero-shot image classification, captioning, and text-to-image retrieval. These embeddings can be used directly off-the-shelf, through probing with shallow networks or adapted via fine-tuning and parameter-efficient methods such as LoRA Hu et al. (2022). Their strong semantic structure makes them attractive candidates for hashing in large-scale retrieval.

**Hashing.** Early hashing approaches combined handcrafted features (e.g., raw pixels, color histograms, edge descriptors) with simple hash functions such as random projections Johnson et al. (1984), PCA Weiss et al. (2008), or iterative quantization Gong et al. (2012). With the rise of pretrained neural networks Deng et al. (2009); Krizhevsky et al. (2012); Simonyan & Zisserman (2014), deep hashing replaced handcrafted features with learned embeddings. Initial approaches applied classical hashing to these features Cao et al. (2017), while subsequent works directly optimized binarization objectives using tanh or sigmoid relaxations, or straight-through estimators to backpropagate through non-differentiable operations Luo et al. (2023). Supervised hashing often uses class labels to sample triplets or construct class-specific hash centers Long et al. (2018); Liu et al. (2018); Hoe et al. (2021); Yuan et al. (2020), whereas unsupervised methods preserve instance-level consistency, sometimes via knowledge distillation from pretrained encoders Luo et al. (2023); Gong et al. (2022); Cao et al. (2023); Ma et al. (2024) or adversarial regularization Cao et al. (2018). Self-supervised learning principles such as contrastive objectives Cao et al. (2023); Qiu et al. (2021); Shen et al. (2024); Jang & Cho (2021), masked patch modeling Shen et al. (2024), and entropy maximization Li & van Gemert (2021) have also been applied to unsupervised hashing. Cross-modal hashing extends these ideas to align codes across modalities such as image–text Li et al. (2024); Liu et al. (2020); Li et al. (2018). A common challenge across these methods is *collapse*, where binary codes degenerate to low-variance solutions; prior work mitigates this with entropy-based regularizers or decorrelation constraints Li & van Gemert (2021); Hoe et al. (2021). Despite these advances, most methods remain paradigm-specific, and rely on multi-term objectives that are complex and difficult to optimize.

**Our distinction.** Unlike prior methods that design separate objectives for different paradigms or rely on complex multi-term losses, we propose **CroVCA** (**Cro**ss-**V**iew **C**ode **A**lignment), a simple principle for learning binary codes. By aligning semantically consistent views with a binary cross-entropy loss and regularizing with coding-rate maximization Wu et al. (2025); Li et al. (2022); Tong et al. (2023), our approach avoids code collapse while promoting balanced, high-entropy codes. This formulation unifies unsupervised and supervised hashing within a single objective, removing the need for paradigm-specific designs. While the same principle can naturally extend to cross-modal settings, our focus in this work is on supervised and unsupervised hashing.

## 3 METHOD

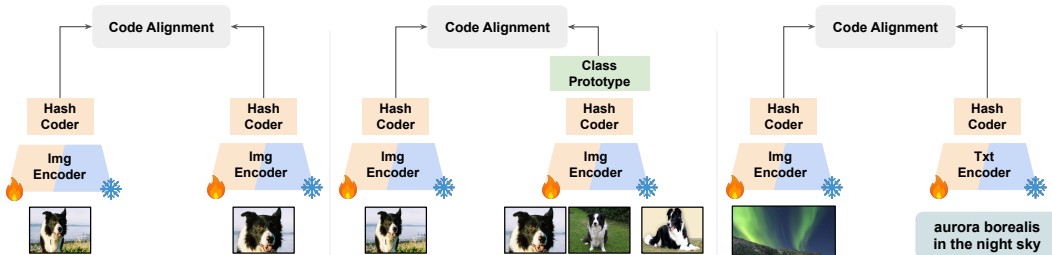

Figure 1: Cross-view code alignment for different hashing setups: unsupervised (left), supervised (middle), and cross-modal (right). Encoders are either frozen or fine-tuned via LoRA.

We introduce **CroVCA** (**Cro**ss-**V**iew **C**ode **A**lignment), a unified, information-theoretic framework for learning compact binary codes on top of foundation model embeddings. Figure 1 summarizes the idea across unsupervised, supervised, and cross-modal settings. We first introduce the problem statement and the HashCoder module, then derive a principled training objective that balances *alignment* (agreement across views) and *diversity* (code utilization).

### 3.1 NOTATION AND PROBLEM STATEMENT

Let $\mathcal{X}$ be the input space (images, text, or both) and $b$ the target hash length. The goal is to learn a mapping
$$\phi : \mathcal{X} \to \{0,1\}^b$$
that preserves semantic similarity, i.e., semantically related inputs should have small Hamming distance.

For each input $x \in \mathcal{X}$ we construct a *paired* example $(x^{(1)}, x^{(2)})$ where the pairing depends on the setting: unsupervised, where two augmentations of the same input are used; supervised, where $x^{(2)}$ is a class-representative (prototype or batch-mean) of $x^{(1)}$'s class; and cross-modal, where paired modalities are used (e.g., image and caption).

Let $y^{(1)}, y^{(2)} \in \{0,1\}^b$ be the observed binary codes for the two views, modeled as realizations of underlying random variables $Y^{(1)}$ and $Y^{(2)}$. The desiderata are: (i) **alignment**: $d_H(y^{(1)}, y^{(2)})$ is small for paired views; and (ii) **diversity**: the marginal distribution of $Y$ should be high-entropy, ensuring balanced and decorrelated bits.

### 3.2 HASHCODER: HASHING NETWORK

We implement $\phi$ as a pretrained encoder $f_\theta$ (frozen or adapted via LoRA) followed by a lightweight MLP, *HashCoder*, that outputs per-bit logits. For a view $x^{(v)}$, $v \in \{1, 2\}$:

$$h^{(v)} = f_\theta(x^{(v)}) \in \mathbb{R}^d, \qquad \text{(backbone embedding)} \tag{1}$$

$$z^{(v)} = \text{HashCoder}(h^{(v)}) \in \mathbb{R}^b, \qquad \text{(logits)} \tag{2}$$

$$p^{(v)} = \sigma(z^{(v)}) \in [0,1]^b, \qquad \text{(bit probabilities)} \tag{3}$$

$$y^{(v)} = \mathbf{1}\{p^{(v)} \geq 0.5\} \in \{0,1\}^b, \qquad \text{(binary code)} \tag{4}$$

where $\sigma$ is the elementwise sigmoid.

**Architecture and design choices.** HashCoder is a compact MLP inspired by SSL projection heads Balestriero et al. (2023). We use two variants: (i) a *large* 3-layer MLP for large datasets, and (ii) a *small* 2-layer MLP for lightweight adaptation. Both variants include a final batch normalization layer, implicitly balancing bit usage, following OrthoHash Hoe et al. (2021). Figure 2 shows the design.

**Training dynamics.** For each paired example, one branch is binarized to serve as the *teacher* ($y^{(1)}$), while the other branch remains soft ($p^{(2)}$) and acts as the *student*. Gradients are stopped on the teacher, so only the student branch is updated. The roles are swapped symmetrically across views, providing discrete supervision without backpropagating through the hard threshold and thus avoiding the need for straight-through estimators Bengio et al. (2013).

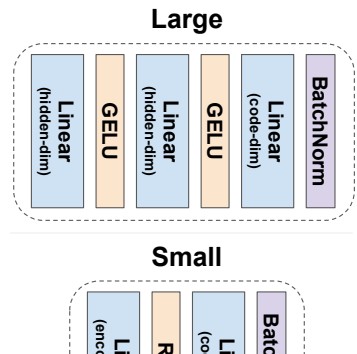

Figure 2: HashCoder design

### 3.3 INFORMATION-THEORETIC OBJECTIVE AND TRACTABLE SURROGATES

We aim to increase the mutual information between codes of paired views:

$$I(Y^{(1)}; Y^{(2)}) = H(Y^{(1)}) - H(Y^{(1)} \mid Y^{(2)}),$$

which decomposes naturally into **alignment** (small conditional entropy) and **diversity** (large marginal entropy). Both terms are intractable for discrete high-dimensional codes; we derive tractable surrogates.

**Conditional entropy $\rightarrow$ binary cross-entropy (alignment).**
Let $P(Y^{(1)} \mid Y^{(2)})$ denote the true conditional distribution and $Q(Y^{(1)} \mid Y^{(2)})$ a surrogate model. By the standard cross-entropy decomposition:

$$H(Y^{(1)} \mid Y^{(2)}) = \mathbb{E}[-\log Q(Y^{(1)} \mid Y^{(2)})] - \mathrm{KL}(P \parallel Q) \leq \mathbb{E}[-\log Q(Y^{(1)} \mid Y^{(2)})],$$

so the conditional entropy is upper-bounded by the expected negative log-likelihood of the surrogate.

We choose $Q$ to be an elementwise-independent Bernoulli distribution parameterized by the soft outputs $p^{(2)}$ of the other branch:

$$Q(Y^{(1)} = y^{(1)} \mid p^{(2)}) = \prod_{j=1}^{b} (p_j^{(2)})^{y_j^{(1)}} (1 - p_j^{(2)})^{1 - y_j^{(1)}}.$$

The negative log-likelihood under this distribution gives exactly the binary cross-entropy (BCE) between the teacher code $y^{(1)}$ and the student probabilities $p^{(2)}$:

$$\mathrm{BCE}(y^{(1)}, p^{(2)}) = -\sum_{j=1}^{b} \left[ y_j^{(1)} \log p_j^{(2)} + (1 - y_j^{(1)}) \log(1 - p_j^{(2)}) \right].$$

Averaging over the batch approximates the expected negative log-likelihood. Symmetrizing across both views yields the alignment loss:

$$\mathcal{L}_{\mathrm{align}} = \frac{1}{2} \left[ \mathrm{BCE}(y^{(1)}, p^{(2)}) + \mathrm{BCE}(y^{(2)}, p^{(1)}) \right].$$

Minimizing $\mathcal{L}_{\mathrm{align}}$ therefore reduces an upper bound on the conditional entropy $H(Y^{(1)} \mid Y^{(2)})$, providing a principled surrogate for alignment.

**Marginal entropy → coding-rate surrogate (diversity).** The marginal entropy,

$$H(Y) = - \sum_{y \in \{0,1\}^b} P(Y = y) \log P(Y = y),$$

promotes balanced, decorrelated bits. Direct computation is infeasible. We define a continuous surrogate using the pre-threshold logits $z$:

$$v_i = \frac{z_i}{\|z_i\|_2}, \quad C = \frac{1}{B} \sum_{i=1}^{B} v_i v_i^\top.$$

Modeling $v_i$ as zero-mean Gaussian, the differential entropy is $h(v) = \frac{1}{2} \log \det(2\pi e \Sigma)$ Ma et al. (2007). Maximizing $\log \det \Sigma$ spreads the vectors along independent directions, increasing diversity after thresholding. The numerically stable *coding rate* surrogate is

$$R(C) = \frac{1}{2} \log \det \left( I + \frac{d}{B} C \right), \quad \mathcal{L}_{\text{div}} = -R(C).$$

The overall hashing objective is

$$\mathcal{L}_{\text{hash}} = \mathcal{L}_{\text{align}} + \lambda \mathcal{L}_{\text{div}},$$

with $\lambda > 0$ controlling the trade-off. Minimizing this loss simultaneously reduces an upper bound on conditional entropy and increases a differentiable surrogate for marginal entropy, yielding a principled framework for unsupervised, supervised, and cross-modal hashing.

# 4 EXPERIMENTS

We evaluate **CroVCA** on standard image retrieval benchmarks, focusing on **supervised** and **unsupervised hashing**. All experiments use either **HashCoder probing** or **LoRA fine-tuning**, with models trained for **5 epochs**. For retrieval, we compute **asymmetric Hamming distance** Jain et al. (2011) between query logits and database codes. Following standard practice, retrieval performance is reported using mean Average Precision (mAP) at typical cutoffs: mAP@1,000 for CIFAR10, ImageNet100, and ImageNet-1k, and mAP@5,000 for FLICKR25K, COCO, and NUS-WIDE. Implementation details are in Table 8 in the Appendix.

## 4.1 TASK-SPECIFIC FINE-TUNING

**Question 4.1.1:** Can foundation model embeddings be efficiently adapted into compact binary codes with lightweight unsupervised fine-tuning?

**Experiment 4.1.1:** We train HashCoder by fine-tuning foundation model embeddings with LoRA at 16, 32, and 64 bits using unsupervised hashing. Retrieval performance is compared with state-of-the-art unsupervised hashing results in Table 1, with additional comparisons in Table 11.

Table 1: Unsupervised hashing of pre-trained foundation models in comparison to state-of-the-art. **Best** and second best results are highlighted.

| Model | CIFAR10 | | | | COCO | | | | FLICKR25K | | | | NUS-WIDE | | | | ImageNet100 | | | |
|---|---|---|---|---|---|---|---|---|---|---|---|---|---|---|---|---|---|---|---|---|
| | Orig | 16 | 32 | 64 | Orig | 16 | 32 | 64 | Orig | 16 | 32 | 64 | Orig | 16 | 32 | 64 | Orig | 16 | 32 | 64 |
| SOTA | – | 94.2 | 95.1 | 95.8 | – | 82.2 | 87.5 | 89.4 | – | 81.8 | **83.8** | 84.9 | – | 81.2 | **83.2** | 84.4 | – | 82.0 | 86.0 | 86.9 |
| **CroVCA (ours)** | | | | | | | | | | | | | | | | | | | | |
| **SimDINOv2** (IN-1k) | | | | | | | | | | | | | | | | | | | | |
| ViT-B | 89.6 | 95.9 | 95.1 | 93.8 | 87.4 | 85.4 | 87.0 | 87.8 | 81.1 | 78.3 | 78.0 | 75.6 | 84.3 | **81.8** | 81.8 | 82.4 | 84.1 | 79.6 | 81.8 | 83.8 |
| **DINOv2** (LVD-142M) | | | | | | | | | | | | | | | | | | | | |
| ViT-B | 95.4 | **98.6** | **98.7** | 97.9 | 88.3 | **87.5** | **89.2** | 89.0 | 76.3 | 69.1 | 69.1 | 68.2 | 79.8 | 75.7 | 77.4 | 77.3 | 88.2 | 87.1 | 88.5 | 89.2 |
| **DINOv3** (LVD-1689M) | | | | | | | | | | | | | | | | | | | | |
| ViT-S | 86.9 | 93.8 | 92.5 | 90.6 | 82.7 | 79.5 | 82.3 | 82.4 | 72.9 | 68.2 | 67.5 | 66.7 | 81.4 | 80.8 | 81.2 | 80.1 | 77.7 | 60.8 | 72.1 | 74.8 |
| ViT-B | 93.6 | 97.7 | 97.7 | 96.7 | 86.6 | 86.7 | 88.8 | 89.1 | 73.0 | 64.2 | 65.3 | 65.4 | 80.7 | 76.7 | 78.9 | 78.9 | 85.9 | 83.5 | 86.1 | 88.9 |
| **DFN** (DFN-2B) | | | | | | | | | | | | | | | | | | | | |
| ViT-B | 94.2 | 93.0 | 93.2 | 93.6 | 87.0 | 83.4 | 87.5 | 89.2 | 83.1 | **83.3** | 82.4 | 81.9 | 84.1 | 80.4 | 83.1 | 83.0 | 81.1 | 61.9 | 73.9 | 78.8 |
| **SWAG** (IG-3.6B → IN1k) | | | | | | | | | | | | | | | | | | | | |
| ViT-B | 89.5 | 92.8 | 91.7 | 90.5 | 88.6 | 84.7 | 89.0 | **90.1** | 79.2 | 82.2 | 80.7 | 78.5 | 82.5 | 80.6 | 82.2 | 82.6 | 94.3 | **91.9** | **93.3** | **94.3** |
| **DeiT** (IN-1k) | | | | | | | | | | | | | | | | | | | | |
| ViT-B | 84.3 | 89.8 | 90.2 | 88.9 | 84.9 | 82.1 | 84.8 | 85.4 | 77.9 | 80.4 | 80.0 | 78.7 | 82.3 | 80.1 | 81.8 | 82.4 | 90.9 | 91.5 | 92.9 | 93.7 |

**Findings 4.1.1:** LoRA fine-tuning with cross-view code alignment consistently matches or surpasses prior unsupervised hashing methods across datasets and bit lengths. Even after only 5 epochs, HashCoder produces competitive or state-of-the-art retrieval performance.

> **Takeaway 4.1.1**
>
> Cross-view code alignment efficiently learns compact binary codes via lightweight LoRA fine-tuning in an unsupervised manner, while preserving class-level semantic structure.

**Question 4.1.2:** How well is semantic structure preserved when embeddings are compressed into very short binary codes via unsupervised hashing?

**Experiment 4.1.2:** To evaluate semantic preservation, we visualize 16-bit HashCoder embeddings on CIFAR10 using t-SNE and compare them with the original 768-dimensional embeddings (Figure 3). Additionally, we perform nearest neighbor retrieval on ImageNet100: for a zebra query, we compare results using the original 768-dim features versus the 16-bit HashCoder codes (Figure 4). In both cases, we start from pretrained SimDINOv2 features and train HashCoder on top via unsupervised hashing using LoRA finetuning of the backbone.

**Findings 4.1.2:** Even with over 40× dimensionality reduction, the 16-bit HashCoder codes preserve the overall class structure. On CIFAR10, t-SNE shows clearly separable clusters corresponding to different categories. On ImageNet100, nearest neighbor retrieval indicates that HashCoder captures semantic features: for a zebra query, it retrieves diverse zebra images, whereas the original embeddings mostly select visually near-identical images. This demonstrates that HashCoder's binary representations retain meaningful semantic organization despite extreme compression.

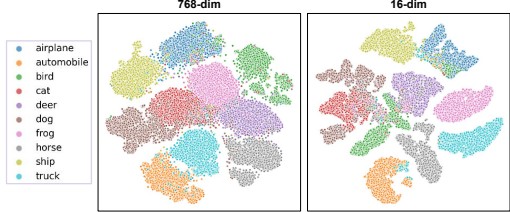

Figure 3: t-SNE of CIFAR10 embeddings: original 768-dim (left) vs. 16-dim HashCoder (right).

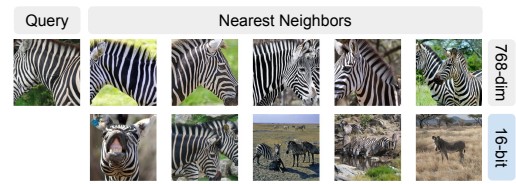

Figure 4: Nearest neighbors of a zebra using SimDINOv2 features vs. HashCoder's 16-bit codes.

> **Takeaway 4.1.2**
>
> Unsupervised cross-view code alignment compresses embeddings into 16-bit codes while preserving class-level semantics, enabling retrieval of diverse and semantically meaningful images.

**Question 4.1.3:** How does cross-view code alignment perform compared to state-of-the-art supervised methods for compact code learning?

**Experiment 4.1.3:** We train HashCoder with class supervision on ImageNet100 and ImageNet-1k, and compare it against FPPQ Liang et al. (2023), the state-of-the-art supervised quantization method, and Ortho-Hash Hoe et al. (2021), the state-of-the-art supervised hashing method (Table 2). Additional comparisons are provided in Table 12. All experiments use 16-, 32-, and 64-bit codes.

Table 2: Supervised hashing.

| Method | Epochs | IN100 16 | IN100 32 | IN100 64 | IN1k 16 | IN1k 32 | IN1k 64 |
|---|---|---|---|---|---|---|---|
| FPPQ (VQ) | 100/90 | 89.5 | 91.2 | 91.5 | 62.0 | 65.4 | 66.4 |
| OrthoHash (Hashing) | 100 | 86.9 | 88.6 | 89.9 | 59.3 | 65.1 | 67.6 |
| Ours (DINOv2) | 5 | 90.2 | 91.1 | 92.1 | 58.4 | 64.3 | 65.9 |

**Findings 4.1.3:** Our approach surpasses FPPQ and OrthoHash on ImageNet100 and achieves competitive performance on ImageNet-1k, despite using only 5 training epochs compared to 90–100

for the other methods. This highlights that cross-view code alignment enables efficient supervised compact code learning.

> **Takeaway 4.1.3**
>
> Cross-view code alignment efficiently produces supervised hash codes while requiring very few training iterations.

**Question 4.1.4:** How does our method perform qualitatively compared to an efficient hashing approach?

**Experiment 4.1.4:** We perform nearest-neighbor retrieval on ImageNet100 for two carefully chosen queries of an indigo bird and a grey langur, to highlight fine-grained distinctions and visual ambiguity. We compare three methods: (1) cosine similarity on the original 768-dimensional SimDINOv2 embeddings, (2) Hashing-Baseline (H-B) Moummad et al. (2025), a fast PCA-based 16-bit hashing method, and (3) HashCoder with 16-bit codes trained via LoRA fine-tuning for 5 epochs on a single GPU in 3 minutes (Figure 5).

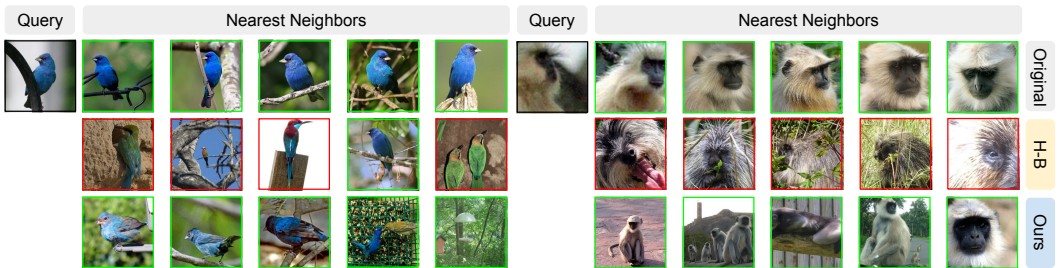

Figure 5: ImageNet100 retrieval results for two queries. Rows: original SimDINOv2 768-dim embeddings; Hashing-Baseline (H-B) Moummad et al. (2025); HashCoder with 16-bit codes trained via unsupervised LoRA fine-tuning.

**Findings 4.1.4:** For the indigo bird query, the original embeddings retrieve visually consistent images of the species. Hashing-Baseline Moummad et al. (2025), although efficient, retrieves only one correct image and four unrelated birds, showing a loss of fine-grained semantic detail. In contrast, HashCoder successfully retrieves all five correct images, including a very small bird (see Figure 9 for a zoom), demonstrating effective preservation of semantic information under 16-bit compression. For the grey langur query, HashCoder retrieves nearest neighbors from the correct class, whereas Hashing-Baseline returns visually similar but semantically incorrect animals. These results highlight that cross-view code alignment maintains semantic structure and is robust to visually challenging or ambiguous queries.

> **Takeaway 4.1.4**
>
> Cross-view code alignment preserves fine-grained class semantics and outperforms fast PCA-based hashing under extreme 16-bit compression, even for ambiguous or visually challenging queries.

## 4.2 TRANSFERABILITY OF IMAGENET FINE-TUNING

**Question 4.2.1:** Can a single HashCoder trained via cross-view code alignment on a large dataset such as ImageNet-1k generate hash codes that transfer effectively to downstream datasets, reducing the need for per-task retraining?

**Experiment 4.2.1:** We train HashCoder on ImageNet-1k via LoRA for multiple bit lengths (16, 32, 64) and evaluate the transfer of these hash codes to CIFAR10, FLICKR25K, COCO, NUS-WIDE, and ImageNet100 without any additional training. Both unsupervised and supervised fine-tuning settings are considered (Table 3).

Table 3: Transferability of hash codes from ImageNet-1k to downstream datasets.

| Method | CIFAR10 | | | | COCO | | | | FLICKR25K | | | | NUS-WIDE | | | | ImageNet100 | | | |
|---|---|---|---|---|---|---|---|---|---|---|---|---|---|---|---|---|---|---|---|---|
| | Orig | 16 | 32 | 64 | Orig | 16 | 32 | 64 | Orig | 16 | 32 | 64 | Orig | 16 | 32 | 64 | Orig | 16 | 32 | 64 |
| **Unsupervised** | | | | | | | | | | | | | | | | | | | | |
| DINOv2 | 95.4 | 93.2 | 95.0 | 95.8 | 88.3 | 82.8 | 85.8 | 86.6 | 76.3 | 76.5 | 78.9 | 78.2 | 79.8 | 74.0 | 76.0 | 77.6 | 88.2 | 79.9 | 84.0 | 85.6 |
| SimDINOv2 | 89.6 | 77.3 | 79.5 | 81.3 | 87.4 | 82.5 | 83.3 | 84.9 | 81.1 | 78.0 | 78.2 | 78.3 | 84.3 | 76.0 | 77.4 | 78.7 | 84.1 | 72.8 | 77.8 | 79.6 |
| **Supervised** | | | | | | | | | | | | | | | | | | | | |
| DINOv2 | 95.4 | 92.9 | 95.2 | 95.7 | 88.3 | 82.2 | 85.8 | 86.7 | 76.3 | 75.4 | 77.0 | 76.7 | 79.8 | 71.6 | 75.2 | 76.4 | 88.2 | 83.0 | 87.8 | 88.3 |
| SimDINOv2 | 89.6 | 78.8 | 82.1 | 82.7 | 87.4 | 79.9 | 83.2 | 84.0 | 81.1 | 76.3 | 76.5 | 76.7 | 84.3 | 73.6 | 76.2 | 76.8 | 84.1 | 74.6 | 81.0 | 83.0 |

**Findings 4.2.1:** HashCoder trained on ImageNet-1k transfers effectively to all downstream datasets, with only minor drops compared to dataset-specific fine-tuning (Table 1). This holds for both unsupervised and supervised hashing, showing that cross-view code alignment produces semantically rich codes that generalize well across datasets.

> **Takeaway 4.2.1**
>
> A single HashCoder trained via cross-view code alignment on a large-scale dataset like ImageNet-1k can generate compact hash codes that transfer effectively to downstream tasks, reducing the need for per-task retraining.

## 4.3 TRANSFERABILITY OF IMAGENET-1K PROBING

Previous experiments demonstrated that LoRA fine-tuning can efficiently produce low-bit hash codes with competitive retrieval performance in just a few training iterations. However, adapting to new datasets typically requires task-specific training, or training a general-purpose HashCoder on a large-scale dataset may incur some performance loss. This motivates investigating whether a single, general-purpose HashCoder, trained once on a large dataset, can generate compact, transferable embeddings that generalize effectively across diverse downstream tasks.

**Question 4.3.1:** Can a single, general-purpose HashCoder, trained via cross-view code alignment on a frozen foundation model using a large-scale dataset, produce compact codes that preserve semantic structure and transfer effectively across different models and downstream tasks?

**Experiment 4.3.1a:** To determine the minimal code length that preserves the semantic richness of the original embeddings, we conduct an ablation study with code lengths ranging from 16 to 256 bits. We use DINOv3's ConvNext-Small (CNX-S) as a frozen backbone and apply HashCoder probing on ImageNet-1k to generate hash codes of different lengths. These codes are then transferred to downstream datasets (CIFAR10, FLICKR25K, COCO, NUS-WIDE, and ImageNet100) to evaluate retrieval performance. Table 4 summarizes the results, allowing us to identify code lengths that match or approach the performance of the full embeddings.

Table 4: Ablation of code dimension with DINOv3 CNX-S using HashCoder probing on IN1k. **Best** and second best results are highlighted.

| Features | CIFAR10 | COCO | FLICKR25K | NUS-WIDE | ImageNet100 |
|---|---|---|---|---|---|
| 768-dim | 94.1 | 86.7 | 75.5 | 78.1 | 88.3 |
| 256-bit | **93.2** | **87.3** | 75.8 | 77.2 | **89.5** |
| 128-bit | 92.9 | 86.6 | 76.0 | **77.3** | 88.4 |
| 64-bit | 92.1 | 85.4 | 77.4 | 76.6 | 87.2 |
| 32-bit | 90.8 | 84.1 | **78.5** | 74.5 | 84.8 |
| 16-bit | 85.9 | 78.6 | 77.3 | 72.2 | 80.7 |

**Findings 4.3.1a:** Retrieval performance generally improves with longer codes, with 256-bit codes achieving the best balance across datasets, closely matching or surpassing the original embeddings. Shorter codes (16–64 bits) still maintain reasonable performance, demonstrating that compact hash codes preserve substantial semantic information.

**Experiment 4.3.1b:** Using the 256-bit codes from the ablation, we perform HashCoder probing on ImageNet-1k with ViT-L, ViT-B, and CNX-S backbones under both unsupervised and supervised settings. The resulting codes are transferred to downstream datasets to evaluate transferability (Table 5).

Table 5: Transfer learning of HashCoder probing on IN1k. Retrieval is evaluated across multiple datasets using original and hashed 256-bit codes.

| Model | CIFAR10 | | COCO | | FLICKR25K | | NUS-WIDE | | ImageNet100 | |
|---|---|---|---|---|---|---|---|---|---|---|
| | Orig | Code | Orig | Code | Orig | Code | Orig | Code | Orig | Code |
| **Random HashCoder** | | | | | | | | | | |
| ViT-B | 94.2 | 87.4 | 86.4 | 74.5 | 73.5 | 62.2 | 81.2 | 68.7 | 85.6 | 77.7 |
| **Unsupervised** | | | | | | | | | | |
| ViT-L | 96.9 | 97.4 | 86.9 | 88.3 | 73.3 | 76.1 | 79.8 | 78.3 | 90.2 | 92.3 |
| ViT-B | 94.2 | 94.8 | 86.4 | 88.4 | 73.5 | 78.5 | 81.2 | 80.2 | 85.6 | 87.6 |
| CNX-S | 94.1 | 93.2 | 86.7 | 87.3 | 75.5 | 75.8 | 78.1 | 77.2 | 88.3 | 89.5 |
| **Supervised** | | | | | | | | | | |
| ViT-L | 96.9 | 97.4 | 86.9 | 87.5 | 73.3 | 76.3 | 79.7 | 78.7 | 90.2 | 93.5 |
| ViT-B | 94.2 | 94.6 | 86.4 | 87.8 | 73.5 | 77.0 | 81.2 | 78.9 | 85.6 | 89.7 |
| CNX-S | 94.1 | 93.1 | 86.7 | 86.6 | 75.5 | 74.6 | 78.1 | 76.6 | 88.3 | 91.3 |

**Findings 4.3.1b:** Across supervised and unsupervised settings, 256-bit codes obtained via Hash-Coder probing preserve the semantic information of the original embeddings. Retrieval performance on downstream datasets is comparable to full embeddings, demonstrating that HashCoder can generate compact, transferable codes efficiently from a single large-scale dataset.

> **Takeaway 4.3.1**
>
> Cross-view code alignment enables frozen foundation models to produce compact hash codes that preserve semantics—creating lightweight, transferable hashing networks without any backbone retraining.

## 5 CONCLUSION

We introduced **CroVCA** (**Cro**ss-**V**iew **C**ode **A**lignment), a simple and efficient framework for adapting foundation models to hashing. It leverages a lightweight hashing network, *HashCoder*, trained either by probing frozen embeddings or via LoRA fine-tuning, supporting both supervised and unsupervised settings. By aligning views through maximization of mutual information, our method produces compact binary codes that preserve the semantic structure of the original embeddings with minimal training. Even extremely low-bit codes capture meaningful class-level information in a fully unsupervised manner. Future work will extend CroVCA to fine-grained retrieval and explore transferability to new domains.

LLM USAGE

We used the GPT-5 model to refine sentence structure and improve grammar while preserving the original meaning. All generated text was carefully reviewed to ensure semantic fidelity and to avoid errors or hallucinated content. The use of large language models is fully disclosed in accordance with the ICLR 2026 policies.

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

# A APPENDIX

## A.1 REPRODUCIBILITY

To facilitate reproducibility, we will release the code for cross-view code alignment at:

`https://anonymous.4open.science/r/cross-view-code-alignment/`

## A.2 TEXT-IMAGE HASHING

While the main paper focuses on applying **CroVCA** to unsupervised and supervised image hashing, we investigate its use for text-image hashing (see Figure 1, right).

### A.2.1 LIGHTWEIGHT FINE-TUNING

**Objective:** Can cross-view code alignment train a hashing network on top of a pretrained text-image foundation model to achieve comparable retrieval performance in low-bit regimes (e.g., 16-bit codes)?

**Experiment:** We perform LoRA fine-tuning on the DFN-Base Fang et al. (2023) text and image encoders, using COCO2017 and FLICKR25K following the evaluation protocol of DDBH Qin et al. (2025). Retrieval performance is measured using mean Average Precision (mAP) for both text-to-image (T2I) and image-to-text (I2T) tasks.

Table 6: Text-image hashing (16-bit codes) with cross-modal retrieval.

| Method | Epochs | Flickr25K | | NUSWIDE | |
|---|---|---|---|---|---|
| | | T2I | I2T | T2I | I2T |
| DDBH (SOTA) | 100 | 82.4 | 84.5 | 71.9 | 70.4 |
| DFN (Base) | - | 66.9 | 64.9 | 48.2 | 46.4 |
| Ours (DFN Hashed) | 5 | 64.8 | 65.0 | 49.9 | 49.6 |

**Findings:** Cross-view code alignment allows rapid adaptation of large text-image models for low-bit hashing. In fewer than 5 epochs ($=2$ minutes), our method compresses 512-dimensional embeddings into 16-bit Hamming codes while preserving a substantial fraction of retrieval performance.

While the results do not surpass state-of-the-art DDBH, the key takeaway is that *cross-view code alignment provides an efficient, low-cost mechanism to learn compact, cross-modal binary codes*. Extending to fully competitive text-image hashing is left as future work.

### A.2.2 PROBING

In Section 4.3, we explored probing of image foundation models for hashing by training on ImageNet-1k and transferring to downstream datasets. Here, we extend this exploration to image-text models.

**Objective:** Can compact 256-bit codes preserve cross-modal semantic alignment between text and image embeddings, enabling effective text-to-image (T2I) and image-to-text (I2T) retrieval?

**Experiment:** We perform MLP probing on DFN text and image encoders (Base and Large) using the CC3M training set to learn 256-bit hash codes. Retrieval performance is evaluated on COCO (Figure 6) and ImageNet-1k (Table 7) under multiple similarity measures: Asymmetric Hamming (AH); Binary cross-entropy (BCE); Symmetric BCE (symBCE)—requires logits from database items. We report both T2I and I2T retrieval using mean Average Precision (mAP@k).

**Results on COCO:** Hashed embeddings capture cross-modal alignment reasonably well, though performance lags behind the original continuous embeddings. BCE outperforms asymmetric Hamming, and symBCE provides a small additional improvement by utilizing full logits from the database.

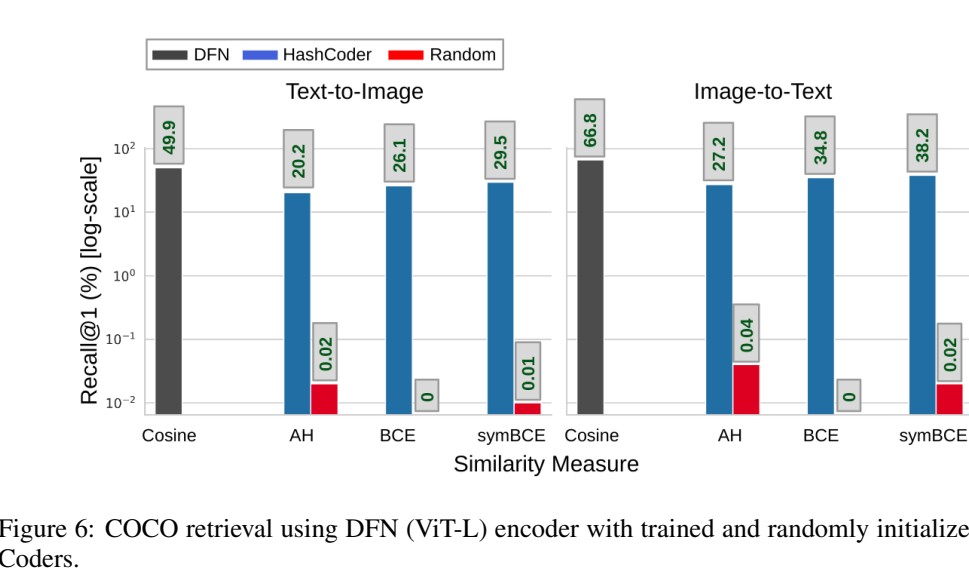

Figure 6: COCO retrieval using DFN (ViT-L) encoder with trained and randomly initialized Hash-Coders.

Table 7: Text-to-Image retrieval performance of DFN models on ImageNet-1k validation set using original features (Cosine) and 256-bit hashed codes via MLP probing on CC3M. Evaluation metrics: mAP@k.

| Model | Original | | | Code (256-bits) | | | | | | | | |
|---|---|---|---|---|---|---|---|---|---|---|---|---|
| | Cosine | | | AH | | | BCE | | | symBCE | | |
| | 1 | 5 | 10 | 1 | 5 | 10 | 1 | 5 | 10 | 1 | 5 | 10 |
| DFN (Base) | 85.0 | 87.8 | 86.3 | 74.4 | 79.0 | 76.0 | 80.4 | 83.9 | 81.1 | 82.3 | 85.1 | 82.8 |
| DFN (Large) | 87.5 | 89.6 | 88.6 | 73.9 | 78.5 | 75.5 | 81.2 | 84.4 | 82.3 | 81.6 | 85.9 | 83.9 |

**Results on ImageNet-1k:** The 256-bit hashed embeddings achieve good retrieval performance. Symmetric BCE consistently improves over BCE, which in turn outperforms asymmetric Hamming, across all $k$ (Figure 7).

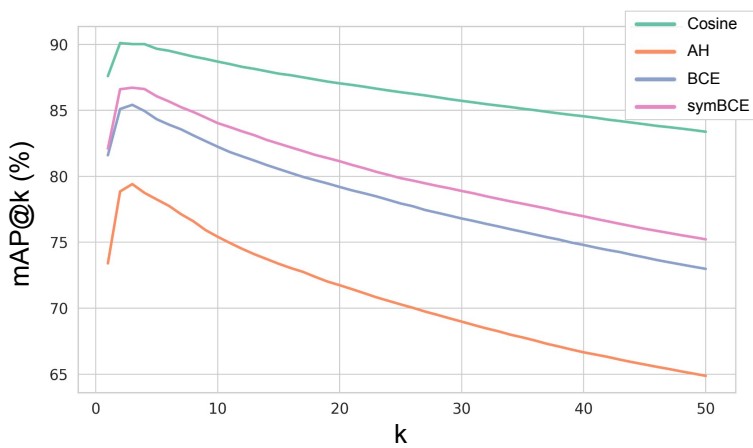

Figure 7: IN1k: text-to-image retrieval using DFN (Large) showing mAP@k for $k \in [1, 50]$.

Figure 8 illustrates retrieval performance on Flickr30K using DFN (Large).

Unlike lightweight fine-tuning, probing text-image models is more challenging: compressing the shared embedding space into a compact Hamming representation is difficult. We hypothesize that this is partly due to modality mismatches in the original embeddings. An investigation of this phenomenon is left for future work.

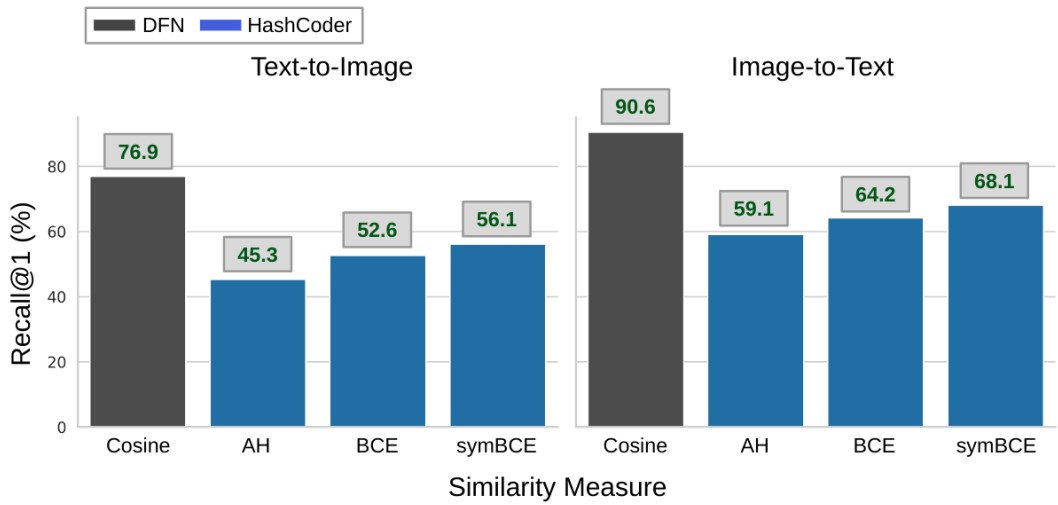

Figure 8: FLICKR30k retrieval using DFN (Large).

Our hashing training relies on a binary cross-entropy objective to align paired views. While effective for finding the closest match, this formulation does not enforce properties such as triangle inequality, unlike Hamming distance or cosine similarity. Exploring its role as a retrieval measure, or incorporating it as a ranking loss to better structure the learned space, may open new avenues for information retrieval.

## A.3 HYPERPARAMETERS AND TRAINING PROCEDURES

Table 8 summarizes the hyperparameter settings used across different training protocols.

Table 8: Hyperparameter settings for different training protocols. LoRA rank is 16 and LoRA dropout is 0.1 for all experiments.

| Hyperparameter | Image | | Image-Text | |
|---|---|---|---|---|
| | Small Datasets | ImageNet-1k | Small Datasets | CC3M |
| Batch size | 256 | 256 | 256 | 256 |
| Optimizer | AdamW | AdamW | AdamW | AdamW |
| Learning rate | 1e-3 | 1e-4 | 1e-3 | 1e-4 |
| Weight decay | 1e-2 | 1e-4 | 1e-2 | 1e-4 |
| Lambda | 0.1 | 0.1 | 0.1 | 1 |
| Image size | 224 | 224 | 224 | 224 |
| Crop size | 40% | 40% | 40% | - |
| # of views (incl. text) | 2 | 2 | 2 | 2 |
| HashCoder hidden layers | 2 | 3 | 2 | 3 |
| Epochs | 5 | 5 | 5 | 1 |
| LoRA rank | 16 | 16 | 16 | - |
| LoRA dropout | 0.1 | 0.1 | 0.1 | - |

## A.4 DATASET DETAILS

Table 9 provides an overview of the datasets used in our experiments.

## A.5 BACKBONE ARCHITECTURES AND CHECKPOINTS

We employed several vision transformer–based backbones in our experiments. Table 10 summarizes the model families, their variants, and pretraining datasets. For SWAG Singh et al. (2022)

Table 9: Dataset statistics for vision-only and vision-language retrieval tasks. For cross-modal datasets, both text-to-image (T2I) and image-to-text (I2T) tasks are considered.

| Dataset | Task Type | Train | Database | Query | Eval Metric |
|---|---|---|---|---|---|
| *Vision-only* | | | | | |
| Flickr25K | Image-to-Image | 4,000 | 20,000 | 1,000 | mAP@5k |
| NUS-WIDE-21 | Image-to-Image | 10,500 | 193,734 | 2,000 | mAP@5k |
| COCO | Image-to-Image | 10,000 | 117,218 | 5,000 | mAP@5k |
| CIFAR10 | Image-to-Image | 50,000 | 50,000 | 10,000 | mAP@1k |
| ImageNet100 | Image-to-Image | 13,000 | 128,503 | 5,000 | mAP@1k |
| ImageNet1K | Image-to-Image | 1,281,167 | 45,000 | 5,000 | mAP@1k |
| *Vision-Language* | | | | | |
| Flickr25K | I2T / T2I | 10,000 | 10,000 | 5,000 | mAP@all |
| NUS-WIDE-21 | I2T / T2I | 10,000 | 193,734 | 5,000 | mAPall |
| Flickr30K | I2T / T2I | 30,000 | 158,915 | 31,783 | Recall@1 |
| COCO | I2T / T2I | 25,000 | 25,000 | 5,000 | Recall@1 |
| ImageNet1K | T2I | - | 50,000 | 1,000 | mAP@[1-50] |

and DeiT Touvron et al. (2021), we used the official Torchvision checkpoints.[1] For SimDINOv2, we used the checkpoint provided in its official repository,[2] for DINOv2 we relied on the official GitHub release,[3] for DINOv3 on the Hugging Face collection,[4] and for DFN on the Hugging Face checkpoint.[5]

Table 10: Backbone models used in our experiments, with variants and pretraining datasets.

| Model Family / Reference | Variants | Training Dataset |
|---|---|---|
| SimDINOv2 | ViT-Base | IN1k |
| DINOv2 | ViT-Base | LVD-142M |
| DINOv3 | CNX-S, ViT-B, ViT-L | LVD-1689M |
| DFN | ViT-Base, ViT-Large | DFN-2B |
| SWAG | ViT-Base | IG-3.6B → IN1k |
| DeiT | ViT-Base | IN1k |

## A.6 FULL COMPARISON WITH STATE-OF-THE-ART

Table 11 provides a more detailed comparison of lightweight task-specific finetuning of unsupervised hashing with state-of-the-art-methods. Our hashing protocol provides the best results overall.

Table 12 provides a more detailed comparison of lightweight task-specific finetuning of supervised hashing with state-of-the-art-methods.

## A.7 NEAREST NEIGHBORS

Figure 9 illustrates an example of nearest-neighbor retrieval for an indigo bird in ImageNet100 using 16-bit hashed representations derived from SimDINOv2 features. Interestingly, the retrieved neighbor suggests that the model captures contextual cues and semantic information beyond low-level visual similarity. This allows the correct identification of the bird's class despite both the heavy compression and the small size of the bird within the image.

---

[1] https://docs.pytorch.org/vision/main/models/generated/torchvision.models.vit_b_16.html

[2] https://github.com/RobinWu218/SimDINO/tree/main

[3] https://github.com/facebookresearch/dinov2

[4] https://huggingface.co/collections/facebook/dinov3-68924841bd6b561778e31009

[5] https://huggingface.co/apple/DFN2B-CLIP-ViT-B-16

Table 11: Unsupervised hashing - comparison with state-of-the-art methods.

| Model | CIFAR10 | | | COCO | | | FLICKR25K | | | NUS-WIDE | | | ImageNet100 | | |
|---|---|---|---|---|---|---|---|---|---|---|---|---|---|---|---|
| | 16 | 32 | 64 | 16 | 32 | 64 | 16 | 32 | 64 | 16 | 32 | 64 | 16 | 32 | 64 |
| **SOTA** | | | | | | | | | | | | | | | |
| IPHASH[(Gong et al., 2022)] | 94.2 | 95.1 | 95.8 | 82.6 | 87.5 | 89.4 | - | - | - | 79.7 | 81.6 | 82.6 | - | - | - |
| HARR*[(Ma et al., 2024)] | - | - | - | 74.8 | 78.9 | 81.6 | 81.8 | 83.0 | 83.8 | 80.7 | 82.6 | 84.1 | - | - | - |
| FSCH[(Cao et al., 2023)] | 87.6 | 91.2 | 92.6 | 76.0 | 78.7 | 79.9 | 81.5 | **83.8** | **84.9** | 81.2 | **83.2** | **84.4** | - | - | - |
| CTMIH[(Shen et al., 2024)] | - | - | - | 80.9 | 83.4 | 84.6 | - | - | - | 79.5 | 81.6 | 82.6 | 82.0 | 86.0 | 86.9 |
| **Ours** | | | | | | | | | | | | | | | |
| **SimDINOv2** (IN-1k) | | | | | | | | | | | | | | | |
| ViT-B | 95.9 | 95.1 | 93.8 | 85.4 | 87.0 | 87.8 | 78.3 | 78.0 | 75.6 | **81.8** | 81.8 | 82.4 | 79.6 | 81.8 | 83.8 |
| **DINOv2** (LVD-142M) | | | | | | | | | | | | | | | |
| ViT-B | **98.6** | **98.7** | **97.9** | **87.5** | **89.2** | 89.0 | 69.1 | 69.1 | 68.2 | 75.7 | 77.4 | 77.3 | 87.1 | 88.5 | 89.2 |
| **DINOv3** (LVD-1689M) | | | | | | | | | | | | | | | |
| ViT-S | 93.8 | 92.5 | 90.6 | 79.5 | 82.3 | 82.4 | 68.2 | 67.5 | 66.7 | 80.8 | 81.2 | 80.1 | 60.8 | 72.1 | 74.8 |
| ViT-B | 97.7 | 97.7 | 96.7 | 86.7 | 88.8 | 89.1 | 64.2 | 65.3 | 65.4 | 76.7 | 78.9 | 78.9 | 83.5 | 86.1 | 88.9 |
| **DFN** (DFN-2B) | | | | | | | | | | | | | | | |
| ViT-B | 93.0 | 93.2 | 93.6 | 83.4 | 87.5 | 89.2 | **83.3** | 82.4 | 81.9 | 80.4 | 83.1 | 83.0 | 61.9 | 73.9 | 78.8 |
| **SWAG** (IG-3.6B → IN1k) | | | | | | | | | | | | | | | |
| ViT-B | 92.8 | 91.7 | 90.5 | 84.7 | 89.0 | 90.1 | 82.2 | 80.7 | 78.5 | 80.6 | 82.2 | 82.6 | 91.9 | 93.3 | 94.3 |
| **DeiT** (IN-1k) | | | | | | | | | | | | | | | |
| ViT-B | 89.8 | 90.2 | 88.9 | 82.1 | 84.8 | 85.4 | 80.4 | 80.0 | 78.7 | 80.1 | 81.8 | 82.4 | 91.5 | 92.9 | 93.7 |

Table 12: Supervised hashing results on IN100 and IN1k datasets. **Best** and second best results are highlighted.

| Method | Epochs | IN100 | | | IN1k | | |
|---|---|---|---|---|---|---|---|
| | | 16 | 32 | 64 | 16 | 32 | 64 |
| **SOTA** | | | | | | | |
| CSQ[(Yuan et al., 2020)] | 90 | 83.7 | 87.5 | 88.7 | 50.4 | 60.6 | 60.9 |
| GreedyHash[(Su et al., 2018)] | 120 | 85.4 | 87.9 | 88.5 | 54.2 | 58.9 | 59.5 |
| OrthoHash[(Hoe et al., 2021)] | 100 | 86.9 | 88.6 | 89.9 | 59.3 | 65.1 | **67.6** |
| FPPQ[(Liang et al., 2023)] | 100/90 | 89.5 | **91.2** | 91.5 | **62.0** | **65.4** | 66.4 |
| Ours (DINOv2) | 5 | **90.2** | 91.1 | **92.1** | 58.4 | 64.3 | 65.9 |

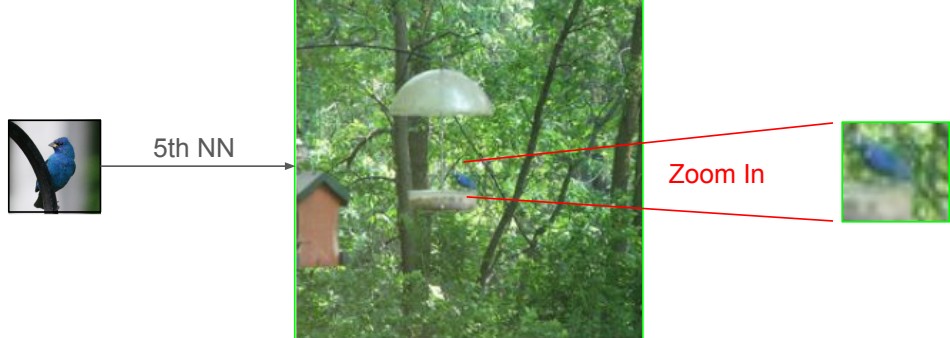

Figure 9: Zoom in on the fifth-nearest neighbor of the query image of a blue bird in ImageNet100 using our 16-bit hashed representations of SimDINOv2.

