# OpenReview forum: "Image Hashing via Cross-View Code Alignment in the Age of Foundation Models"
_ICLR.cc/2026/Conference — ICLR 2026 Conference Withdrawn Submission_

### Official Review · Reviewer_h3gK · 2025-10-28

**Soundness:** 2
**Presentation:** 3
**Contribution:** 2
**Rating:** 4
**Confidence:** 4

**Summary:**

This paper proposes CroVCA for learning binary hash codes using foundation model with a cross-view alignment principle. The method employs a single binary cross-entropy loss for alignment between semantically similar views (augmentations or class prototypes) and a coding-rate regularization term to prevent code collapse. A lightweight HashCoder network is introduced, which can be used either as a probe on frozen embeddings or adapted via LoRA fine-tuning. The empirical studies on supervised, unsupervised, and cross-modal settings verify the effectiveness of the proposed method.

**Strengths:**

- This work proposes an unified and simple formulation that uses a single binary cross-entropy loss for cross-view alignment, combined with coding-rate regularization that can be used for unsupervised, supervised, and cross-modal settings.

- The experiments on multiple datasets verifies the effectiveness of the proposed method. The ablation studies, t-SNE visualizations, and qualitative retrieval improve the empirical claims.

**Weaknesses:**

- While this work claims that they propose a unified framework to deal with unsupervised, supervised, cross-modal setting. However, the idea of using augmentation, class prototype has been widely used. In addition, the losses used here, including cross-entropy loss and coding-rate are similarly used in some works. This framework does not sufficiently give a specific theoretical or algorithmic contribution in hashing literature, and it is more likely a simple and efficient unification.

- I still concern why this work is superior than existing works. The ideas of augmentation, prototype, and losses are commonly used in many other works.

- The setup of experiments should be in more detail. It should be justified whether the comparisons with baselines are fair. with the same backbone, training set, and evaluation metrics. I notice that some results in some tables e.g., Table 11 are missing. Do you directly adopt these results from their papers?

- It is suggested to compare more sota baselines in each setting for verification.

**Questions:**

Please address the concerns in Weakness

---

### Official Review · Reviewer_ukDX · 2025-10-30

**Soundness:** 2
**Presentation:** 4
**Contribution:** 1
**Rating:** 2
**Confidence:** 5

**Summary:**

The paper presents a lightweight MLP-based image hashing network to learn the binary codes based on the output features of visual foundation models. Specifically, the cross-entropy loss is exploited for alignment between different views of the same input image. The code-rate maximization is further leveraged to regulate the hash code learning to alleviate code collapse and enhance the code balance with diversity. The learned hashing code can be employed as the probing head on the frozen embeddings or to adapt encoders via LoRA fine-tuning. Good training efficiency and performances have been achieved on various image datasets with different visual foundation models.

**Strengths:**

1.	Multi-view image alignment learning is a reasonable manner for the unsupervised hashing. The lightweight design is effective to learn compact hash codes on the features of visual foundation models.
2.	The paper is well-written and the figures are very clear.
3.	Performance gains are attained on different foundation models and the transferability for zero-shot hash code generation is also verified.
4.	The ablation studies are comprehensive to demonstrate the effectiveness of the proposal.

**Weaknesses:**

1.	Although the experiments and evaluations are comprehensive, the technical contribution of the proposal is still very limited. The design of the MLP hashing network is very common in the old-school deep hashing works. The multi-view learning is also the typical solution in unsupervised learning, such as the contrastive learning. The learning objective function for code balance can be interpreted as the orthogonal constraint across different samples, which is similar to the work of OrthoHash and some other previously traditional deep hashing works (even though most works do not mention it as a contribution but a normal regularization term).
2.	The improvements are contributed by the strong semantic features extracted from the visual foundation models. The comparisons are not fair to the SOTA methods.
3.	The performances are almost saturated on the traditional datasets, e.g., CIFAR10 and ImageNet. Is there any evaluation on a much bigger scenario to verify the application for the real-world image retrieval?

**Questions:**

Please see the weaknesses.

---

### Official Review · Reviewer_zs1V · 2025-10-30

**Soundness:** 3
**Presentation:** 3
**Contribution:** 2
**Rating:** 4
**Confidence:** 3

**Summary:**

The paper proposes CroVCA, a unified hashing objective built upon foundation-model embeddings. The setup utilises a lightweight HashCoder MLP head (available in small/large variants) with a final BatchNorm layer to balance bit usage; it can probe frozen encoders or adapt them efficiently via LoRA fine-tuning. Training forms paired views (augmentations, class prototypes, or cross-modal pairs), aligns them with binary cross-entropy (BCE) while preventing collapse by maximizing a coding-rate surrogate that encourages high-entropy, decorrelated bits. Practically, training uses a teacher-student, stop-gradient scheme that avoids straight-through estimators. Experiments span unsupervised and supervised image hashing (5-epoch protocol) with standard mAP evaluations and report notable speed (e.g., COCO unsupervised in <2 minutes; ImageNet100 supervised in ~3 minutes on a single GPU). The appendix also explores cross-modal hashing, acknowledging it does not yet surpass SOTA but highlights rapid low-bit adaptation.

**Strengths:**

- This paper decomposes mutual information into alignment and diversity and replaces intractable terms with BCE (upper-bounding conditional entropy) and a log-det coding-rate term on normalized logits to promote balanced, decorrelated bits-simple to implement and grounded in information-theoretic identities.
- The stop-gradient teacher-student scheme supplies discrete supervision without STEs, reducing engineering friction while aligning codes across views.  ￼
- A 5-epoch recipe with a small head (or LoRA) yields strong results and very fast wall-clock times on a single GPU.

**Weaknesses:**

- The diversity term hinges on modeling normalized logits as zero-mean Gaussian to connect entropy to a log-det objective R(C)=\tfrac12\log\det(I+\tfrac{d}{B}C). While this is a common relaxation, the paper does not provide a direct analysis tying this particular surrogate-versus alternatives (e.g., other entropy proxies or orthogonality penalties) to the observed gains after hard thresholding. Moreover, because the final BatchNorm also balances bits, and the total loss is L_{\text{hash}}=L_{\text{align}}+\lambda L_{\text{div}}, the current experiments do not include ablations that isolate: (i) the effect of Ldiv alone; (ii) the effect of the final BatchNorm; or (iii) their interaction. As a result, it remains unclear which mechanism (coding-rate vs BatchNorm vs BCE-only) is chiefly responsible for improvement, a material gap for a method whose novelty rests on this surrogate.

- Several cross-modal experiments use BCE/symmetric-BCE as the retrieval score, and the authors themselves note BCE does not enforce triangle inequality, unlike Hamming or cosine similarity. This raises interpretability/fairness concerns when comparing hashed retrieval against cosine or Hamming baselines; a clearer rationale and consistent ranking metric would strengthen claims.

- The paper reports minute-level training times on a single GPU but, in the sections that report them, does not accompany these with detailed wall-clock logs or variance/throughput analysis, which limits how confidently others can extrapolate the claimed efficiency.

- In text-image hashing, the method currently lags SOTA (e.g., DDBH) at 16 bits; while the paper frames this as a speed/efficiency trade-off and future work, it tempers the generality of the “unified” claim for multimodal retrieval at present.

**Questions:**

Please check above.

---

### Official Review · Reviewer_2nED · 2025-11-01

**Soundness:** 2
**Presentation:** 2
**Contribution:** 1
**Rating:** 4
**Confidence:** 2

**Summary:**

This paper proposes CroVCA (Cross-View Code Alignment), a unified principle for learning compact and discriminative binary codes to address the computational inefficiency of high-dimensional embedding-based large-scale image retrieval. It addresses limitations of existing hashing methods—such as complex pipelines, multi-term objectives, single-paradigm specialization, and long training times—by using a single binary cross-entropy loss to enforce consistency across semantically aligned views and coding-rate maximization as a regularizer to prevent code collapse. The paper also designs HashCoder, a lightweight MLP network with a final batch normalization layer for balanced codes, which can function as a probing head on frozen foundation model embeddings or adapt encoders via LoRA fine-tuning. Experimental results show CroVCA achieves state-of-the-art performance across benchmarks with only 5 training epochs; for example, it completes 16-bit unsupervised hashing on COCO in under 2 minutes and supervised hashing on ImageNet100 in ~3 minutes on a single GPU, demonstrating its efficiency, adaptability, and broad applicability.

**Strengths:**

The paper’s key strengths lie in its targeted solution to a critical inefficiency in large-scale image retrieval: it proposes a conceptually simple and unified framework (CroVCA) that avoids the complex pipelines and multi-term objectives of existing hashing methods, uses a lightweight MLP-based HashCoder, compatible with both frozen embeddings and LoRA fine-tuning, for flexibility, and delivers impressive empirical efficiency—achieving state-of-the-art results across benchmarks in just 5 training epochs and completing fast 16-bit hashing tasks on a single GPU, which effectively addresses the high computational cost of foundation model-based retrieval.

**Weaknesses:**

The paper lacks in-depth analysis of its core "cross-view" component, such as how view generation is stabilized or how it performs when view quality degrades; it provides limited evaluation of HashCoder’s adaptability across more diverse foundation model backbones, leaving uncertainty about its broad applicability; and it offers insufficient discussion of performance tradeoffs in extreme low-bit scenarios or comparisons to recent lightweight hashing methods that also target efficiency, which weakens the rigor of its claimed advantages.

**Questions:**

How you ensure the stability and quality of view generation—especially in cases where input images have ambiguous semantics or limited texture information?
Could you explain why coding-rate maximization is superior in this framework, and provide ablation studies showing how the removal of this regularizer affects code diversity and retrieval performance on noisy datasets?

---

### Note · Authors · 2025-11-14

I have read and agree with the venue's withdrawal policy on behalf of myself and my co-authors.